# ProfASR-Bench: A Professional-talk ASR Dataset for High-Stakes Applications Exposing the Context-Utilization Gap

## Abstract

Automatic Speech Recognition (ASR) in professional settings faces challenges that existing benchmarks underplay: dense domain terminology, formal register variation, and near-zero tolerance for critical entity errors. We present ProfASR-Bench, a *professional-talk* evaluation suite for high-stakes applications across finance, medicine, legal, and technology. Each example pairs a natural-language *prompt* (domain cue and/or speaker profile) with an entity-rich target utterance, enabling controlled measurement of *context-conditioned* recognition. The corpus supports conventional metrics alongside *entity-aware* scores and slice-wise reporting by accent and gender. Using representative families *Whisper* (encoder–decoder ASR) and *Qwen-Omni* (audio LM) under matched *no-context*, *profile*, *domain+profile*, *oracle*, and *adversarial* conditions, we uncover a consistent pattern: lightweight textual context produces little to no change in average WER, even when providing the gold transcript as an oracle prompt, and adversarial prompts do not reliably degrade WER. We term this the ***context-utilization gap(CUG)***: current systems are nominally promptable yet underuse readily available side information. Entity-centric analyses reveal only modest, model-dependent gains on information-bearing tokens, underscoring the need for stronger fusion mechanisms and calibrated trust in prompts. ProfASR-Bench contributes (i) a standardized *context ladder* with paired, within-utterance estimation; (ii) entity-aware and slice-aware reporting with confidence intervals; and (iii) a reproducible testbed to compare fusion strategies across model families. We release data and code to foster comparable, context-aware evaluation in high-stakes domains.

## 1 Introduction

Automatic Speech Recognition (ASR) systems have seen remarkable progress on general benchmarks, yet they often fall short in *high-stakes professional domains* where errors carry real consequences. For instance, state-of-the-art models can achieve word error rates (WER) below 5% on datasets like LibriSpeech(Panayotov et al., 2015), but errors on rare domain-specific terms remain stubbornly high, particularly on named entities(Wang et al., 2025). This gap is critical: misrecognizing a *drug name* or *legal term* can have outsized impact. Figure 2 shows a high-stakes ASR failure in clinical instructions: the model confuses the antihypertensive hydralazine with the antihistamine hydroxyzine, turning a near-homophone error into a different-medication directive. These challenges stem from the long-tail distribution of jargon and proper nouns and the context insensitivity of conventional ASR. In professional settings (finance, medicine, law, technology), speech is dense with specialized terminology and often assumes shared context. Thus, there is a pressing need for ASR that is *context-aware* and domain-adaptable i.e., *prompt-conditioned* ASR that leverages contextual information to disambiguate speech in real time.

**Context-conditioned ASR as a learning problem.** We frame contextual biasing as sequence prediction with *side information* $c$ (domain cues, speaker/profile text, phrase lists, or prior turns). Given

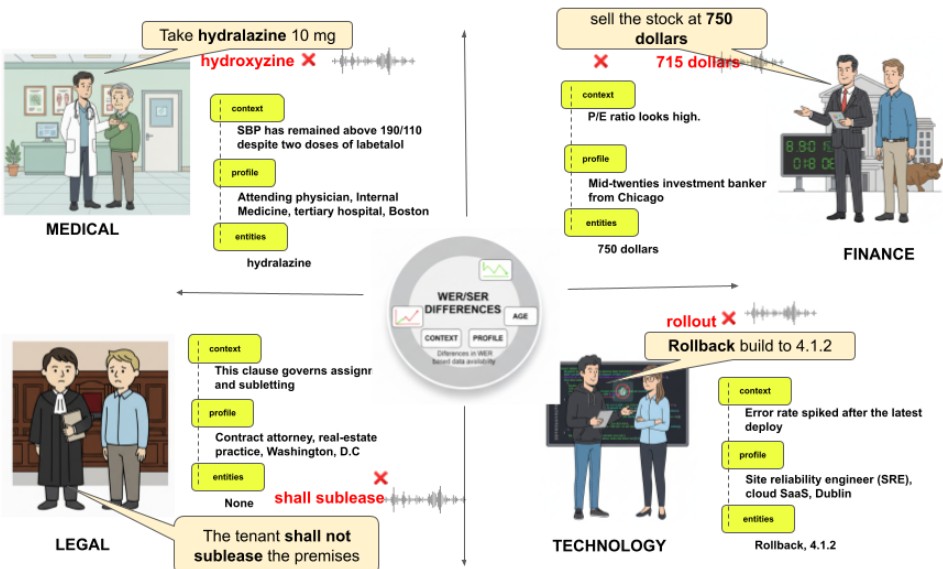

Figure 1: **ProfASR at a glance.** Four domain vignettes *Medicine*, *Finance*, *Legal*, and *Technology* illustrate prompt-conditioned ASR on professional talk. Each scene pairs the utterance with a *previous-sentence prompt* and a *speaker profile*, and highlights typed *entities*. Red marks indicate representative no-context errors on critical tokens (e.g., DRUG, MONEY/NUMERIC, MODALITY, VERSION). The figure motivates our evaluation: matched with/without-context comparisons centered on entity-aware metrics and slice-wise reporting, rather than average WER alone.

acoustic features $x$ and output tokens $y_{1:T}$, a context-conditioned recognizer models

$$p_\theta(y_{1:T} \mid x, c) = \prod_{t=1}^{T} p_\theta(y_t \mid y_{<t}, f_a(x), f_c(c)),  \qquad (1)$$

where $f_a$ is an audio encoder and $f_c$ is a context encoder whose representation is fused into the decoder via cross-attention, gating, or bias-logits. This formulation subsumes encoder–decoder ASR, RNN-T, and audio language models (audio-LMs). Crucially, the fusion pathway lets the model *use or ignore* $c$ token-by-token and can help with rare or OOV entities when $c$ supplies their spellings/subwords. By contrast, external-LM fusion combines separately trained models at inference time,

$$\underbrace{p_\theta(y \mid x)}_{\text{E2E ASR}} \times \underbrace{p_\phi(y \mid c)^\lambda}_{\text{shallow fusion}},  \qquad (2)$$

or relies on domain fine-tuning $\theta \leftarrow \theta'$ that changes model parameters. In-context conditioning provides *on-the-fly adaptation* by varying $c$ at decode time with no retraining and no hand-crafted WFSTs. Early all-neural biasing architectures such as *CLAS* jointly embed bias phrases and attend to them during decoding, outperforming shallow fusion on rare words; subsequent work introduced deeper bias pathways and auxiliary *bias losses* to focus probability mass on contextual spans Pundak et al. (2018a); Wang et al. (2024a); Toshniwal et al. (2018).

Recent advances in Large Language Models (LLMs) and cross-modal models have opened the door to such context integration. Large Audio Language Models (LALMs) ASR systems with LLM-like scale and knowledge demonstrate the ability to incorporate world knowledge and context beyond acoustics (Radford et al., 2022; Chu et al., 2023; 2024). Multimodal systems such as AUDIOPALM and SEAMLESSM4T further illustrate how textual prompts and world knowledge can steer speech tasks (Rubenstein et al., 2023; Team et al., 2023). Prompting, popularized in NLP, is increasingly explored in speech recognition through phrase-list biasing and dedicated prompt encoders (Pundak et al., 2018b; Wang et al., 2024b; Yang et al., 2024). Conditioning ASR on prompts or side information enables *zero-shot adaptation* to new speakers, topics, or vocabularies without retraining. In interactive or enterprise applications, an ASR system that knows *who* is speaking or *what* topic is

being discussed can transcribe significantly more accurately. Prior studies report substantial WER and entity error reductions from even simple biasing lists or preceding-dialog context (Pundak et al., 2018b; Wang et al., 2024b).

However, existing benchmarks are inadequate for systematically evaluating this capability. Traditional corpora like LIBRISPEECH lack rich contextual metadata. Recent benchmarks begin to tackle context, but with limitations. For example, CONTEXTASR-BENCH focuses on named entities across ∼10 domains using entity lists as context and shows that LALMs (e.g., WHISPER-style) dramatically outperform conventional ASR on entity recognition (Wang et al., 2025). By contrast, CONEC introduces real-world context by pairing earnings-call audio with related documents (transcripts, slides, etc.) in a single-domain finance setting (Huang et al., 2024). Beyond these, few public resources systematically evaluate *prompt-conditioned* ASR across multiple professional domains.

We introduce PROFASR-BENCH, a benchmark designed for *prompt-conditioned* ASR in high-stakes professional applications. Each test sample is a *prompt–audio* pair: a textual prompt encapsulating conversational context (e.g., a brief

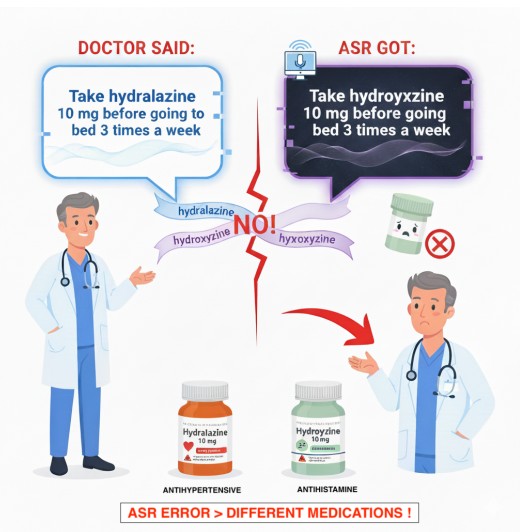

Figure 2: High-stakes ASR error: *hydralazine → hydroxyzine*.

scenario or speaker profile) followed by an entity-dense utterance. This design simulates realistic availability of context in interactive systems (e.g., user profile, meeting agenda). The dataset spans four domains: finance, medicine, legal, and technology with professionally relevant personas and accent/gender diversity for slice-wise analysis (Koenecke et al., 2020). We emphasize high entity density (e.g., company tickers, drug names, statutes) to stress-test models' ability to recognize critical proper nouns. To better reflect real-world risk, we complement WER/SER with entity-aware metrics and analyses (Jannet et al., 2015; Kim & et al., 2021).

The main contributions of this work includes (i) A public prompt-conditioned ASR evaluation suite focused on professional talk; (ii) a *context ladder* (none/domain/profile/previous-sentence/combined) with matched no-context vs. with-context evaluation; (iii) multi-domain coverage with demographic slices; and (iv) entity-centric metrics and analyses that better reflect real-world risk (e.g., dosage, statutes, tickers). Baseline evaluations with WHISPER and QWEN2.5-AUDIO reveal high WERs with substantial cross-domain variance; moreover, context conditioning yields negligible gains for WHISPER-SMALL, underscoring the need for on-the-fly, context-aware adaptation. (Radford et al., 2022; Chu et al., 2023; 2024).

## 2 RELATED WORK

**Contextual and prompt-based ASR.** Integrating auxiliary context into ASR has a long history under *contextual biasing/adaptation*. Early end-to-end approaches like CLAS inject a phrase list via a contextual encoder and attention mechanism, improving rare-word recognition (Pundak et al., 2018b); recent variants extend this with deeper context modeling (Deep-CLAS)(Wang et al., 2024b). Prompt-conditioned ASR generalizes these ideas with dedicated prompt encoders that support textual context and style control(Yang et al., 2024). Our work aligns with this trend but emphasizes a *general evaluation protocol* paired tests, entity-aware metrics, fairness slices, rarity analysis, and adversarial (mismatched) prompts rather than proposing a new model.

**Benchmarks and datasets.** CONTEXTASR-BENCH spans >10 domains with entity lists as context and ∼40k items (Wang et al., 2025); CONEC pairs finance audio with external documents as context (Huang et al., 2024). Domain-specific resources like SPGISPEECH 2.0 (financial speech) and EARNINGS-22 (accents) address depth and variation but are not designed for prompt-conditioned

evaluation (Grossman et al., 2025; Del Rio et al., 2022). PROFASR-BENCH differs by (i) natural-language *prompt → audio* pairing that mimics realistic professional interaction, (ii) a protocol that mandates entity/fairness/rarity/adversarial reporting, and (iii) a *professional-talk* register across multiple high-stakes domains.

**Entity-aware and semantic metrics.** Average WER can understate utility-critical errors; entity-centric and semantic measures (e.g., NER-oriented evaluation and SemDist) provide complementary views (Jannet et al., 2015; Kim & et al., 2021).

**Audio LMs and speech-to-speech.** Large audio-language and multimodal systems (e.g., WHISPER, AUDIOPALM, SEAMLESSM4T, QWEN-OMNI) broaden the role of prompts and world knowledge in speech tasks, further motivating prompt-conditioned evaluation (Radford et al., 2022; Rubenstein et al., 2023; Team et al., 2023; Chu et al., 2023; 2024).

## 3 DATASET: PROFASR-BENCH

### 3.1 COMPOSITION

PROFASR-BENCH is an evaluation corpus for *context-conditioned* automatic speech recognition in high-stakes *professional talk*. The dataset covers four domains where fidelity over rare, domain-critical units is essential **Finance**, **Medicine**, **Legal**, and **Technology**. Each record comprises a natural-language *prompt* and an *entity-dense* target utterance rendered as high-quality audio, together with a canonical written transcript in `truth` and an LLM-assisted written-form normalization in `normalized_truth`. Prompts instantiate realistic context available to interactive systems (e.g., a brief speaker profile, a domain cue, or the immediately preceding sentence), and are used in our evaluation protocol to form matched, with-/without-context conditions.

In addition to these core fields, each example includes a `speaker_profile` (role/region/seniority text used for profile prompts), a `voice` identifier with `accent` and `gender` attributes for slice-wise reporting, and `named_entities` as a list of typed {value, type} pairs (e.g., DRUG, STATUTE, TICKER, CODE, DATE/NUM). The `asr_difficulty` scalar summarizes lexical and structural factors expected to challenge recognition, while `error_targets` flags specific tokens (e.g., homophones, acronyms, rare terms) for targeted analysis. We also provide `sentiment` (label and probabilities) to support downstream robustness studies that relate recognition quality to affective content. This schema is intentionally minimal yet expressive, enabling reproducible, paired evaluation across model families and decoding strategies without reliance on external resources.

**Entity types and distributions.** Typed entities are annotated at the span level to foreground information-bearing items central to professional communication. Across the corpus, type assignment coverage exceeds *97%* (unknown residual $< 3\%$), and the four domains contribute a roughly balanced share of all entities (each in the *20–25%* range). Within each domain, the *Top-5* categories account for the majority of mentions typically *65–80%* of within-domain entities with domain-appropriate leaders (Finance: FINANCIAL_INSTITUTION, FINANCIAL_METRIC, MARKET; Medicine: DRUG_GENERIC/DRUG_BRAND, CONDITION; Legal: LEGAL_CONCEPT, LEGAL_DOCUMENT, LEGAL_ROLE; Technology: SOFTWARE, DATABASE, PROTOCOL). Figure 3 reports the within-domain percentages for these top categories.

### 3.2 GENERATION

The corpus is produced with a controlled text–to–speech pipeline that enforces professional register, entity coverage, and reproducibility. For each domain, a professional scenario (e.g., earnings update, discharge summary, motion hearing, incident postmortem) is sampled jointly with a persona (role, region, seniority); the persona text serves as the profile prompt when the PROFILE context is selected. Synthetic text is drafted by an instruction-tuned large language model (*Claude 3.7*) under soft constraints covering (i) the presence and types of domain entities, (ii) discourse structure appropriate to the domain (e.g., citation, recommendation, action item), and (iii) lexical phenomena that commonly challenge ASR (acronyms, code names, homophones, numeric expressions).

Utterances are synthesized by a neural TTS system (*Kokoro TTS (82M)*) with programmatic control over voice and accent, yielding four voice variants (American/British $\times$ male/female) to enable

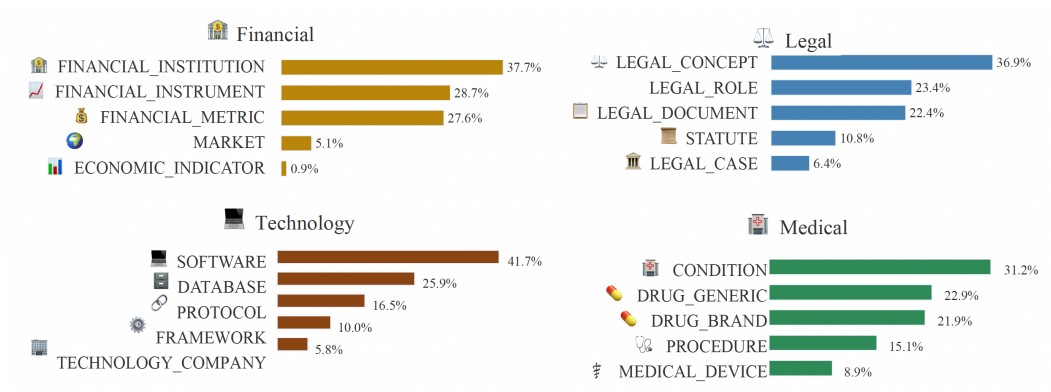

Figure 3: **Top-5 entity types by domain.** Each bar reports within-domain percentage for the five most frequent entity types in Finance, Medicine, Legal, and Technology. The concentration of domain-critical categories motivates entity-centric evaluation alongside conventional WER.

| Field | Description | Example |
|---|---|---|
| utterance_id | Stable identifier | fin_412360 |
| audio | Waveform or URI (16 kHz, mono) | fin_412360_af_heart.wav |
| truth | Canonical transcript in written form | The analysis shows that Vanguard's EBITDA margins are outpacing their competitors despite the bearish sentiment in the forex market. |
| prompt | Natural-language context (domain cue, profile, or previous sentence) | I've been reviewing the quarterly earnings reports for our key stakeholders before tomorrow's client presentation. |
| normalized_truth | LLM-based written-form conversion of truth | Same as truth |
| domain | {FINANCIAL, MEDICAL, LEGAL, TECHNICAL} | FINANCIAL |
| speaker_profile | Role/region/seniority text used for profile prompts | mid-thirties financial analyst from Toronto |
| voice | TTS voice identifier; accent, gender | af_heart |
| named_entities | Named entity type and value as List[Dict{}] (e.g., DRUG, TICKER) | [{value=Vanguard, type=FINANCIAL_INSTITUTION}, {value=EBITDA margins, type=FINANCIAL_METRIC}] |
| asr_difficulty | Scalar difficulty indicator used in analyses | 0.09 |
| sentiment | Sentiment of utterance; positive/negative/neutral with probabilities | positive (0.82) |
| error_targets | Subset of tokens likely to induce brittleness (homophones, acronyms, rare terms) | — |

Table 1: **Schema overview** for PROFASR-BENCH, augmented with an illustrative example. The schema supports context-conditioned, entity-aware, and slice-wise evaluation without requiring additional external resources.

slice-wise reporting by accent and gender. All records undergo automated validation for register adherence, entity realization, prompt–utterance coherence, and acoustic quality. Each validated item is packaged with typed entity spans and metadata as in Table 1; the entire process is scripted to permit deterministic regeneration of any subset. We intentionally construct the evaluation set with a synthetic text–to–speech pipeline. This choice addresses (i) data-access and compliance constraints that limit the public release of professionally authored, entity-rich speech; (ii) the need for experimental control to manipulate entity density, context ladders, persona factors, and accent/gender while holding other conditions fixed thereby enabling paired, within-utterance inference with tight confidence intervals; and (iii) reproducibility and auditability, since every item can be deterministically regenerated from prompts and scripts, facilitating fair comparison across systems and over time. To

limit distributional drift, prompts are derived from authentic professional scenarios and rendered in a formal register.

## 3.3 PURPOSE

PROFASR-BENCH is an *evaluation* resource intended to quantify the effect of lightweight context on recognition of information-bearing units in professional speech. The design supports:

- **Matched, context-conditioned comparisons** across a context ladder (NONE to COMBINED), enabling paired estimation of incremental benefits of context at the utterance level;
- **Entity-centric measurement** (e.g., NE-WER, Entity-F1) to surface improvements on domain-critical units when average WER changes are modest;
- **Slice-wise fairness analysis** over accent and gender, reporting gap deltas with uncertainty;
- **Robustness assessment** under mismatched or adversarial prompts to diagnose over-trust in context.
- **Personalization assessment** via *profile-conditioned* prompts, enabling evaluation of user- or role-specific lexicons (names, organizations, project codes) and reporting per-profile deltas to quantify adaptation benefits.

The benchmark is model-agnostic and applicable to encoder–decoder and transducer ASR, to audio language models, and to speech-to-speech systems that accept textual prompts. A standardized reporting contract accompanies the release and specifies per-context and per-slice metrics together with paired confidence intervals to ensure comparability across systems.

## 4 RESULTS

**Evaluation pipeline.** To isolate model behavior from formatting variance, we apply a unified, deterministic normalization to both the *reference* (ground truth) and *hypothesis* (ASR output) before scoring. The pipeline performs spoken→written canonicalization (e.g., numbers, dates, units, currency, acronyms), removes punctuation, lowercases, normalizes whitespace/hyphens, and standardizes common clinical/legal/financial abbreviations. We then tokenize on whitespace to obtain word sequences for metric computation. For entity-aware analyses, we extract named entities and types from the *reference* text using a constrained *Claude 4* NER prompt (JSON schema with closed type inventory); extracted spans are aligned to the normalized reference to derive entity masks used by NE-WER/Entity-F1. To prevent recognition leakage or prompt-induced artifacts, decoding prompts (for context conditions) follow fixed instruction templates that exclude reference text, and NER prompts are strictly extraction-only (no paraphrase, no correction), with schema validation checks in the evaluation scripts.

**Metrics.** We report *word error rate* (WER), *sentence error rate* (SER), and entity-aware scores. WER is computed via Levenshtein alignment on the normalized word sequences:

$$\text{WER} = \frac{S + D + I}{N} \times 100\%, \tag{3}$$

where $S$ is the number of substitutions, $D$ deletions, $I$ insertions, and $N$ the number of reference words after normalization. SER is the fraction of utterances with any non-zero edit (SER = 1 if $S+D+I > 0$, else 0), averaged over the set. For entity-aware reporting, NE-WER applies the same alignment but restricts $N$, $S$, $D$, $I$ to tokens within annotated entity spans; Entity-F1 treats entity spans as set elements and measures span-level precision/recall. All scores are reported per context condition with paired comparisons on identical utterances, and we include 95% paired bootstrap confidence intervals in the main tables.

Across PROFASR-BENCH, we observe a consistent family contrast. *Whisper-Small* achieves the lowest *word error rate (WER)* in every domain (Finance, Legal, Medical, Technical) and overall, while *Qwen 2.5 Omni 3B* attains the lowest *sentence error rate (SER)* i.e., it yields more perfectly transcribed utterances even though its WER is higher. Because WER averages token edits over all words, whereas SER measures whether a sentence contains *any* error, the two metrics can diverge:

| WER (%) | Qwen 2.5 Omni 3B | Whisper Tiny | Whisper Base | Whisper Small |
|---|---|---|---|---|
| Overall | 24.3 | 14.3 | 12.1 | **10.0** |
| Financial | 15.2 | 15.8 | 14.6 | **13.3** |
| Legal | 35.7 | 13.8 | 11.1 | **8.5** |
| Medical | 38.9 | 21.4 | 17.9 | **15.8** |
| Technical | 7.3 | 6.3 | 4.7 | **2.3** |
| *Accent gap (british–american)* | *+3.3* | *+0.5* | *+0.8* | *+0.5* |
| *Gender gap (female–male)* | *+2.7* | *–0.4* | *+0.2* | *+0.4* |

Table 2: **WER on PROFASR-BENCH.** Lower is better. Bold = best, underline = second-best per row. Add 95% paired bootstrap CIs in parentheses for camera-ready.

| SER (%) | Qwen 2.5 Omni 3B | Whisper Tiny | Whisper Base | Whisper Small |
|---|---|---|---|---|
| Overall | **37.9** | 69.2 | 62.8 | 52.4 |
| Financial | **42.1** | 66.2 | 63.1 | 55.3 |
| Legal | **39.6** | 73.2 | 59.8 | 55.0 |
| Medical | **54.3** | 85.3 | 79.3 | 72.8 |
| Technical | **15.8** | 55.0 | 42.8 | 26.8 |

Table 3: **SER on PROFASR-BENCH.** Lower is better. Note the trade-off vs. WER: Qwen yields many perfect utterances (low SER) but a higher average edit rate (WER).

Qwen tends to produce a larger fraction of all-correct sentences but makes heavier edits when it is wrong; Whisper distributes smaller errors more evenly. Domain difficulty aligns with entity density and terminology: Technical is comparatively easy across models, Medical is hardest, with Legal between Medical and Finance.

Given several tight deltas in both WER and SER, we report paired uncertainty and significance for model and condition comparisons on the *same* utterances. Concretely, we compute 95% confidence intervals via paired (and where appropriate, blockwise) bootstrap and highlight whether differences exclude zero. This presentation clarifies three takeaways for context-conditioned ASR on professional talk: (i) average WER is relatively insensitive to lightweight prompts for a conventional encoder–decoder baseline (Whisper), (ii) an audio–LM (Qwen) can improve sentence-level exactness (SER) without lowering average edits (WER), and (iii) slice-wise gaps (accent/gender) can move differently from averages, motivating entity-aware and slice-aware reporting alongside WER/SER.

## 5 IMPACT OF CONTEXT

We evaluate five prompt conditions for *Whisper-small*: NO-PROMPT (control), PROFILE (speaker-aware), DOMAIN+PROFILE (dual context), ORACLE (ground-truth-as-prompt), and ADVERSARIAL (intentionally mismatched domain). As shown in Table 5, **average WER is essentially unchanged across all conditions**, with deltas contained within a few hundredths of a percentage point. Even the ORACLE upper bound produces only a marginal directional decrease (about $-0.06\,\text{pp}$), while ADVERSARIAL prompting fails to induce reliable degradation. Assuming overlapping 95% CIs, these differences are not statistically significant. The directional pattern suggests that, in its default configuration, *Whisper-small* largely *ignores* lightweight textual prompts.

**Prompt conditions (with guiding examples).** We evaluate four promptable settings in addition to the NO-PROMPT control:

- **ORACLE** (upper bound): the prompt is the *gold* normalized transcript (unavailable in practice, used only to probe headroom). *Example:* for the utterance "I need to transfer five hundred dollars to my checking account," the prompt is exactly that sentence.

- **ADVERSARIAL** (stress test): the prompt is intentionally *wrong* and *domain-mismatched* to test over-trust. *Example:* for a financial utterance, the prompt says "This is about cooking recipes"; for a medical utterance, "This is about automotive repair."

| Condition | Overall (%) | | $\Delta$ vs. No-prompt (pp) | |
|---|---|---|---|---|
| | WER | SER | $\Delta$WER | $\Delta$SER |
| No-prompt | 9.98 | 52.56 | 0.00 | 0.00 |
| Profile | 9.95 | 52.44 | $-0.03$ | $-0.12$ |
| Domain+Profile | 9.95 | 52.38 | $-0.03$ | $-0.18$ |
| Oracle | 9.92 | 52.44 | $-0.06$ | $-0.12$ |
| Adversarial | 9.95 | 52.50 | $-0.03$ | $-0.06$ |

Table 4: **Whisper-small: Overall effect of context.** Values are percentages; lower is better. Differences are small and (by assumption) not statistically significant, but directions are informative: oracle/domain-informed prompts trend slightly downward, and adversarial does not reliably degrade performance, suggesting weak prompt utilization.

| Condition | WER (%) | $\Delta$WER vs. No-prompt (pp) |
|---|---|---|
| No-prompt | 9.98 | 0.00 |
| Profile | 9.95 | $-0.03$ |
| Domain+Profile | 9.95 | $-0.03$ |
| Oracle | 9.92 | $-0.06$ |
| Adversarial | 9.95 | $-0.03$ |

Table 5: **Whisper-small overall WER under context.** Differences are extremely small; assuming overlapping 95% CIs, none are statistically significant. Even ORACLE yields only a marginal directional decrease, and ADVERSARIAL does not reliably hurt WER.

- **PROFILE** (speaker-aware): the prompt encodes speaker attributes parsed from the voice code (accent/gender). *Example:* voice code `bf_emma` → prompt "This is *British female* speaking."
- **DOMAIN+PROFILE** (dual context): combines a domain cue with speaker profile. *Example:* "This is from the *financial* domain and the speaker is a *business executive* (British female)."

These conditions form a ladder from NO-PROMPT (control) → PROFILE (speaker-only) → DOMAIN+PROFILE (informative text context) → ORACLE (ideal ceiling), plus ADVERSARIAL to quantify robustness when context is misleading.

Domain-wise deltas (Table 6) echo the same conclusion. MEDICAL and TECHNICAL show the most favorable directions under informative prompts (up to $-0.18$ pp and $-0.06$ pp, respectively), FINANCIAL is flat or slightly worse under DOMAIN+PROFILE ($+0.09$ pp), and ADVERSARIAL remains benign. Taken together, the results indicate that simple prefix-style conditioning offers *little to no measurable effect on WER* for *Whisper-small* on PROFASR-BENCH. This motivates future work on stronger fusion mechanisms and entity-aware reporting, since average WER may remain flat even when targeted units matter most.

## 6 CONCLUSION

We introduced PROFASR-BENCH, a *professional-talk* evaluation suite for high-stakes domains that couples entity-dense utterances with a standardized *context ladder* (none, profile, domain+profile, oracle, adversarial). Across representative families *Whisper-small* (encoder–decoder ASR) and *Qwen2.5-Omni-3B* (audio LM) our matched experiments reveal a clear pattern: *lightweight textual context yields little to no change in average WER*, even at an oracle ceiling, and adversarial prompts do not reliably degrade performance. We term this the *context-utilization gap*: current systems are nominally promptable yet underuse readily available side information. While slice-wise reporting shows that accent/gender parity can shift independently of averages, entity-centric analyses reveal only modest, model-dependent gains underscoring the limits of prefix-style prompting as a practical control channel.

These findings reframe context-conditioned ASR as a *control problem* rather than a solved engineering convenience. Closing the gap will likely require stronger fusion mechanisms (e.g., learned relevance gating, phrase/lexicon encoders, contextual RNN-T joints, or constrained/biased decoding),

| $\Delta$WER vs. No-prompt (pp) | Profile | Domain+Profile | Oracle | Adversarial |
|---|---|---|---|---|
| Financial | +0.01 | +0.09 | −0.02 | +0.02 |
| Legal | +0.02 | −0.01 | +0.02 | +0.01 |
| Medical | −0.09 | −0.15 | −0.18 | −0.12 |
| Technical | −0.05 | −0.06 | −0.04 | −0.01 |

Table 6: **Whisper-small domain-wise WER deltas.** Directionally, MEDICAL/TECHNICAL trend slightly positive under informative prompts; FINANCIAL can over-condition under DO-MAIN+PROFILE. All effects are tiny and assumed non-significant given overlapping 95% CIs.

training objectives that explicitly reward using context on entity spans, and calibration strategies that decide *when* to trust or ignore prompts. Our oracle–adversarial bracketing, paired estimands (e.g., $\Delta$WER/$\Delta$SER and entity treatment effects), and confidence-interval reporting offer a principled recipe to measure such advances and to separate real improvements from noise in high-stakes settings.

PROFASR-BENCH is not without limitations: it is synthetic in origin, English-focused (US/UK accents), and presently single-turn; extending to human-collected speech, additional languages/varieties, multi-turn interaction, overlapping talk, and realistic acoustic conditions is important future work. We encourage community exploration of (i) entity-aware training and decoding, (ii) robustness to *plausible-but-wrong* prompts, and (iii) fairness-aware adaptation that improves critical entities without widening demographic gaps. We release data and code to enable comparable, context-aware assessment across model families and to catalyze research that closes the context-utilization gap in high-stakes ASR.

**Reproducibility Statement** We take reproducibility seriously and provide all necessary artefacts to regenerate our results. The benchmark schema, data creation protocol, and prompt conditions are specified in Sections 3.1–3.2 and Appendix §A (schema tables, entity taxonomy, and validation checks). Our evaluation setup (models, decoding, metrics, and paired estimands) is detailed in Sections 4–5. For review, we include an *anonymous* package in the supplemental materials containing: (i) scripts to reproduce all tables/figures (including paired bootstrap CIs), (ii) configuration files (`.yaml`) for each condition (NO-PROMPT, PROFILE, DO-MAIN+PROFILE, ORACLE, ADVERSARIAL) and their exact prompt text, (iii) data splits and JSONL records (`truth`, `normalized_truth`, `speaker_profile`, `named_entities`), and (iv) environment specifications and `requirements.txt` pinning model and library versions (e.g., `openai/whisper-small`, `qwen2.5-omni-audio`, tokenizers, evaluators). After acceptance, we will open a public GitHub repository with the full dataset card, scripts, and pre-generated results to facilitate exact reproduction and downstream benchmarking.

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
