# OpenReview forum: "ProfASR-Bench: A Professional-talk ASR Dataset for High-Stakes Applications Exposing the Context-Utilization Gap"
_ICLR.cc/2026/Conference — Submitted to ICLR 2026_

### Official Review · Reviewer_ADWg · 2025-10-28

**Soundness:** 2
**Presentation:** 2
**Contribution:** 2
**Rating:** 4
**Confidence:** 3

**Summary:**

This paper proposes a benchmark to evaluate context-dependent ASR capabilities in professional domains. The benchmark is designed to assess the model's understanding of given situations within context and to analyze the impact of contextual information on ASR performance. This study utilized Whisper and Qwen-omni models for experiments and found that both demonstrated low performance on the proposed benchmark.

**Strengths:**

- This paper proposes a benchmark for prompt-conditioned ASR in professional talk settings. By pairing audio with prompts, the study clearly defines the task focus.
- The comparison between oracle and no-prompt experiments yields intriguing results. These findings highlight a limitation in models claiming context grounding ASR, offering significant insights for the research field.
- This paper constructed and released data aligned with the task of transcribing expert-level terminology. It identified four high-stakes professional domains and developed specific datasets for each.

**Weaknesses:**

- I believe that the necessity for this benchmark is not convincingly justified in the benchmark design choice.
    - While the domains are divided, there is no implication beyond their individual construction. A more detailed analysis on the reason of performance decrease or domain-specific issues for each domain could clarify the purpose of constructing the four distinct domains.
    - In the current setup, existing benchmarks appear sufficient for validation. It seems feasible to assess differences in performance with or without context using existing ASR benchmarks, questioning the need for this specific benchmark.
    - The examples in Figure 1 suggest that errors in Finance, Technology, and Legal domains are not domain-specific but represent conventional error cases that could occur in any speech. The errors proposed by the authors appear fundamentally similar to transcription errors occurring in the general domain; rather their significance depending on the domain is merely amplified.
    - Additionally, most data is synthesized and annotated automatically (section3.2), yet there is insufficient evidence to confirm alignment with professional ASR standards.

- Data generation process (section3.2) lacks numerous details.
    - The authors mention that a professional scenario is sampled (lines 208-210), but the source is unclear. Which specific data were utilized?
    - How were prompts designed? The text mentions a brief speaker profile, a domain cue, or the immediately preceding sentence (line 184), yet lacks detail on the selection criteria and types.
    - Regarding the dataset, how were asr_difficulty, error_targets, and sentiment constructed? Were annotations model-based? If so, what process was used for tagging? The explanation of traits introduced in lines 186-194 is insufficient.
    - Most critically, the automated validation for entity-dense TTS (lines 261-262) requires clarification. It is essential to specify the validation process, particularly the method for audio-level entity annotation and its validity. How is the pronunciation accuracy of specific professional entities verified automatically?
    - Overall, the explanations are overly descriptive yet omit crucial details. For instance, section 3.1, lines 263-270, and section 3.3 seem more suited to an appendix. I believe that enhancing detail is more high-stake at this stage.

- The analysis of the experiment seems overly generalized.
    - Currently, Section 5 might be more accurately titled "Impact of Context on Whisper-Small" rather than just "Impact of Context." The experiments involving only the Whisper-Small model do not sufficiently support the conclusion that ASR models inadequately incorporate context. To verify the "impact of context," experiments should involve a wider range of models, including those specifically designed for context dependency.
    - Whisper, in particular, does not fully account for context-dependent ASR, given its training objectives. Therefore, I believe that it is predictable that providing naive prompts, as shown in Table 1, does not yield effective results. Using models like Qwen-omni, which are designed with context consideration from the outset, seems more appropriate in this section.
    - The benchmark's goal is to evaluate ASR at a high level, yet the experiments in this paper focus primarily on small-sized models. This limitation represents a significant constraint of the study. While the content intended to be demonstrated in Section 5 appears compelling, including experiments with a broader range of large models would enhance persuasiveness.

**Questions:**

- The paper contains several vague terms (significantly / accurately / systematically / addresses / facilitating / …) and lacks detailed explanations. I believe thorough proofreading would be necessary.
- (line 183-185) Could the authors provide specific examples of prompts, such as a brief speaker profile, a domain cue, or the immediately preceding sentence?
    - When using verifiable prompts like the immediately preceding sentence, metrics like the inclusion of verifiable phrases (e.g., accuracy) can be used for evaluation. I view that relying solely on WER for evaluation may result in a superficial level of analysis.
    - This issue is related to Weakness#1.
- Is there a specific reason for using only the Kokoro TTS model for data generation? Considering its small size of 82M, how is the reliability of its audio ensured? (pronunciation / tone / etc..)

---

### Official Review · Reviewer_BsZV · 2025-10-31

**Soundness:** 2
**Presentation:** 4
**Contribution:** 3
**Rating:** 2
**Confidence:** 4

**Summary:**

ProfASR-Bench is a new benchmark for ASR in professional domains conditioned on context documents, generated synthetically using text to speech systems. Baselines with a small set of smallish ASR models shows that they fail to utilize context very effectively.

**Strengths:**

The problem is important. Even though ASR is sort of solved for casual situations, when you get into contexts with specialized vocabulary they can often fall apart (and aren't as adaptive as one would hope).

The presentation of the paper is generally good. The related work seems well-written and fair. I'm not aware of a prior work that does something quite like this, but I'm also not super familiar with related work.

There are some interesting experiment variants on mismatched prompts to measure how well the model handles misleading context, etc. It seems clear that these small ASR models don't use context very effectively.

**Weaknesses:**

So as I was reading, I was completely with you until "The corpus is produced with a controlled text–to–speech pipeline". As far as I can tell, this isn't mentioned until you get into the weeds of Section 3.2. (Yes, it's mentioned in the limitations paragraph in the conclusion, but this is something that obviously should be highlighted upfront.)

Obviously real data is preferable, but synthetic benchmarks are still useful to demonstrate that models can't do something. But there are some additional problems.

First, I don't see any section talking about how you know that e.g. domain-specific vocabulary is actually pronounced correctly by the text to speech system. There is a sentence about "All records undergo automated validation" but it doesn't seem to talk about this, and it's not clear what this automated validation means. Another way to validate something like this would be to provide a human baseline, which is also absent. The scores for Whisper Small in Table 2 are not that far from where you might expect a human to be (and who knows where the missing larger Whisper models would be given that they clearly improve with scale...).

Second, like I said, synthetic benchmarks are most useful if the results are surprising, but your model baselines are only Qwen 2.5 Omni 3B and (the smaller sizes of) Whisper. (Which version of Whisper?) Where are state of the art large audio understanding models, and especially ones that have more knowledge in professional domains due to scale, like Gemini, GPT, etc.?

**Questions:**

I'd like to hear more about the "automated validation" mentioned in the paper, and whether/how you know that domain-specific vocabulary is being pronounced correctly. Or any human validation of the outputs, measurements of correctness rate, etc.

Extra baselines with more capable speech recognition models/LLMs could sway my opinion.

I think you should mention the synthetic nature of the dataset in the abstract/intro.

As a minor point, I don't think this really follows: "Closing the gap will likely require stronger fusion mechanisms (e.g., learned relevance gating, phrase/lexicon encoders, contextual RNN-T joints, or constrained/biased decoding),". My understanding is that Whisper was trained with previous text tokens, i.e. the chunk of the transcript prior to the given example. That's a very different kind of context from lexicons, so while they do suggest that you can put lexicons in there, it seems very plausible that you just need to actually train the model to do it with data in the right format, rather than some complicated architectural change.

---

### Official Review · Reviewer_oeWH · 2025-11-03

**Soundness:** 2
**Presentation:** 3
**Contribution:** 2
**Rating:** 4
**Confidence:** 4

**Summary:**

The paper argues that current ASR systems, even modern encoder–decoder models (Whisper) and audio LMs (Qwen-Omni), do not effectively use lightweight textual context in high-stakes, professional domains (finance, medicine, legal, tech), despite being nominally “promptable.” To make this measurable, the authors introduce PROFASR-BENCH, an evaluation suite where each example is a prompt-audio pair: a natural-language prompt plus an entity-dense synthetic professional utterance rendered via TTS, with rich metadata (domain, accent, gender, entity types, difficulty, error targets). The benchmark defines a “context ladder” (no prompt -> profile -> domain+profile -> oracle -> adversarial) and requires entity-aware and slice-aware reporting (accent, gender), not just average WER. Using small Whisper variants and Qwen 2.5 Omni 3B under matched conditions, the paper finds a consistent pattern: WER barely moves across context conditions, even when giving the gold transcript as an oracle prompt, and even adversarial prompts don’t reliably hurt. They call this the context-utilization gap (CUG): current ASR systems have an input channel for context but largely ignore it. The paper positions PROFASR-BENCH as a reproducible testbed to study better fusion, entity-aware training, and robustness to misleading prompts.

**Strengths:**

1) **Clear problem framing:** The paper isolates a very concrete, currently under-measured problem: “ASR systems say they can take context, but do they actually use it?”  That’s crisp and relevant to multimodal work coming out since Whisper. By naming it context-utilization gap, the paper gives the community a handle to talk about it.

2) **Well-structured evaluation protocol:** The “context ladder” (none, profile, domain+profile, oracle, adversarial) is simple, reproducible, and very easy to adopt in other labs. It also lets the community probe both headroom (oracle) and robustness (adversarial) using the same utterance, which is good experimental hygiene. This is, in my view, the paper’s most portable contribution.

3) **Strong reproducibility:** The whole pipeline is synthetic, scripted, and the evaluation scripts (normalization, NER extraction, prompt templates) are promised for release, configuration files for each context condition are listed, models are common (whisper-small, qwen2.5-audio). This makes it relatively easy for others to re-run and to plug in their fusion strategies.

4) **Useful Negative result:** Showing that even an oracle textual prompt doesn’t help Whisper-small beyond ~0.06 pp WER is counter-intuitive and therefore interesting. It pushes the community to look inside the fusion path rather than just “add a prompt.”

**Weaknesses:**

1) **Synthetic-only, TTS-only data for a “professional-talk” claim:** While the paper justifies synthetic generation well (compliance, controllability, entity density), the title and motivation lean heavily on high-stakes, professional talk where acoustics, disfluencies, channel conditions, and speaker behavior matter. Using clean Kokoro TTS with four voice variants, EN-only, is a distributional mismatch from real medical dictations, earnings calls, court hearings, etc. A small human-recorded or noisy subset (even 200–500 utterances) would strengthen the external validity.

2) **Narrow model comparison:** The tables mainly compare whisper-tiny/base/small vs. qwen2.5 omni 3B, observe an interesting WER-SER trade-off, and stop there. I think the benchmark misses other types of models, such a commercial or larger general model, and models trained specific for contextual (that could be a shallow fusion). Without this, the reader is left wondering whether the benchmark is challenging, or just mismatched to the tested models.

3) **No runtime / cost reporting:** Since the paper positions the benchmark as something practitioners should run across context conditions, it would help to report decoding cost / audio length / real-time factor for the baselines, so readers can estimate feasibility.

**Questions:**

- Why the best version of Whisper was not included?
- Can you add one “designed-for-context” baseline?
- Please consider adding some “real-audio” slices, maybe from publicly releasable professional speech (e.g., earnings calls segments), and run the same context ladder. That will reassure readers that the result is not a TTS artifact.

---

### Official Review · Reviewer_QF4P · 2025-11-03

**Soundness:** 2
**Presentation:** 2
**Contribution:** 1
**Rating:** 2
**Confidence:** 4

**Summary:**

This paper introduces ProfASR-Bench, a benchmark for evaluating context-conditioned automatic speech recognition (ASR) in professional domains (finance, medicine, legal, technology). The dataset pairs natural-language prompts with entity-dense utterances, enabling controlled measurement of context-aware recognition. The authors evaluate Whisper and Qwen-Omni models under various context conditions (no-context, profile, domain+profile, oracle, adversarial) and find that lightweight textual context produces minimal WER improvements, even with oracle prompts, a phenomenon they term the "context-utilization gap" (CUG). The benchmark emphasizes entity-aware metrics, slice-wise fairness analysis, and reproducible evaluation.

**Strengths:**

1. **Tackles a meaningful problem**: The hydralazine/hydroxyzine example (Figure 2) is compelling—near-homophone confusions that swap antihypertensive for antihistamine demonstrate real clinical risk. As ASR deployment scales in medicine, legal proceedings, and financial domains, entity-level errors carry outsized consequences that aggregate WER masks. The focus on information-bearing tokens in professional speech addresses a genuine gap between benchmark performance and deployment safety.
2. **Experimental design**: The context hierarchy is good—starting from no context and building up to oracle/adversarial creates a principled way to probe what models actually use. The paired evaluation design is compelling for controlling confounds.
3. **Entity-aware evaluation**: Moving beyond aggregate WER to look at what actually matters (drug names, legal terms, financial figures) is the right lens for high-stakes domains. NE-WER and Entity-F1 make sense here.
4. **Multi-domain scope**: Spanning finance, medicine, legal, and tech provides reasonable breadth across distinct professional vocabularies and entity types, avoiding the limitations of single-domain benchmarks while capturing common challenges in entity-dense technical speech.

**Weaknesses:**

1. **Synthetic data limits real-world validity**: The entire corpus is generated via Claude 3.7 + Kokoro TTS, creating a closed loop of synthetic text and synthetic audio. Real professional speech contains disfluencies, code-switching, environmental noise, and domain-specific prosody that TTS cannot fully replicate. Without any validation against actual clinical recordings, earnings calls, or legal proceedings, claims about "high-stakes applications" lack grounding. At minimum, a small-scale human speech validation set would establish whether findings transfer.

2. **Insufficient baseline coverage**: Testing only Whisper and Qwen-Omni, neither of which is designed for contextual biasing, inherently limits the generalizability of the CUG finding. The related work extensively discusses CLAS, Deep-CLAS, and PromptASR as architectures explicitly built for context integration, yet none are evaluated. Without these natural baselines, it's unclear whether the gap is fundamental or an artifact of testing the wrong models.

3. **Statistical claims unsupported by presentation**: Tables 4-6 report WER deltas of 0.01-0.18 percentage points and make directional interpretations ("MEDICAL trends slightly positive") while acknowledging differences are non-significant with overlapping confidence intervals (CI). The actual CIs are never shown. Either present the error bars and discuss statistical power, or refrain from interpreting non-significant effects as meaningful trends.

4. **Missing cross-benchmark contextualization**: No comparison with ContextASR-Bench or ConEC using the same models makes it impossible to assess whether ProfASR-Bench is harder, easier, or captures different phenomena. Does Whisper-small achieve 10% WER on those benchmarks too? Is the entity density actually higher here? Without this positioning, the contribution relative to existing contextual ASR evaluation remains unclear.

5. **Limited prompt engineering exploration**: The evaluation uses only simple prefix-style prompts (e.g., "This is British female speaking"). Given the central claim about context underutilization, more comprehensive prompt engineering is needed entity lists, instruction-formatted prompts, post-audio context positioning, or multi-turn dialogue simulation. If the goal is to stress-test context usage, trying one prompt format then declaring a "gap" seems premature.

6. **Unvalidated entity annotations**: Named entities are extracted using Claude-4 with schema constraints, creating a synthetic-on-synthetic pipeline where LLM-generated text is annotated by another LLM. No inter-annotator agreement is reported, and no subset undergoes human validation. The paper emphasizes entity-aware evaluation as a key contribution, yet the quality of entity labels the foundation of NE-WER and Entity-F1, remains unverified.

7. **Insufficient demographic coverage for fairness claims**: Four TTS voices (American/British × male/female) cannot support meaningful "slice-wise fairness analysis." Real-world professional speech encompasses far richer accent diversity (South Asian, East Asian, African, etc.), age-related vocal characteristics, and other variations.

8. **Core findings**: The paper's central contribution that context provides negligible WER improvement even with oracle prompts is presented as empirical observation without deeper investigation. Do models attend to the prompts at all? Is the fusion pathway active? Are prompts encoded properly? No attention analysis, ablation studies, or diagnostic experiments are provided. For ICLR, documenting that something doesn't work is valuable when accompanied by insight into why it fails or how to fix it. The conclusion speculates about "stronger fusion mechanisms" and "calibration strategies" but these remain unexplored suggestions rather than tested hypotheses.

**Questions:**

1. Can the authors provide validation comparing synthetic vs. real professional speech performance?
3. Why not test contextual-biasing architectures (CLAS, Deep-CLAS) to determine if CUG is universal?
4. What are the actual 95% CI ranges in Tables 4-6? Should directional claims be removed if non-significant?
5. How do baselines perform on ContextASR-Bench/ConEC for comparison?
6. Have you explored prompt format ablations (position, entity lists, length)?
7. What is entity annotation inter-annotator agreement? Any human validation?
8. For "high-stakes," why not task-specific metrics (medication error rate, legal term accuracy)?

---

### Meta-Review · Area_Chair_AdjP · 2026-01-04

**Summary:**

While the reviewers were appreciative of the motivation and presentation of the new benchmark, which addresses a valid shortcoming of existing benchmarks in the literature, they also raised several issues:
1. The benchmark was generated synthetically (LLM Text + TTS). So the quality of the data, in comparison to real human speech, is unclear. There can be serious issues with pronunciation, useful disfluencies and other issues that come with real data, which is not well covered.
2. Lacking comparisons using widely used techniques and models from the literature. In fact, there are results in the literature that clearly show that having an oracle transcript in context can significantly improve the quality of ASR (e.g., https://ieeexplore.ieee.org/abstract/document/9746406).
3. Missing a deeper analysis as to why the models are not able to use context as well as it should.

**Reviewer Concerns:**

The authors did not submit a rebuttal.

**Reviewer Scores:**

n.a.

---

### Decision · Program_Chairs · 2026-01-26

Reject